# Optimization of Transcription Factor Genetic Circuits

**DOI:** 10.3390/biology11091294

**Published:** 2022-08-31

**Authors:** Steven A. Frank

**Affiliations:** Department of Ecology and Evolutionary Biology, University of California, Irvine, CA 92697, USA; safrank@uci.edu

**Keywords:** genetic regulatory networks, transcription factors, optimization, differential equation models, artificial neural networks, automatic differentiation, Julia programming language

## Abstract

**Simple Summary:**

Transcription factors (TFs) are proteins that bind to DNA and control the expression of genes, including other TF genes. A common challenge in cellular biology is to link the molecular attributes of TF binding to the system level properties of cellular dynamics. This article introduces a computational method to search for how a TF network might achieve particular functions. For example, how could one link the molecular parameters of TF production rates and DNA binding to a TF network that maintains a cellular circadian rhythm in the absence of external light signals? How could such a system also reset its clock when an external light signal is present? By computationally optimizing such models of TF dynamics, one can formulate hypotheses about how cells achieve particular functions. One can also gain insight into the ways in which cells process information through their TF networks.

**Abstract:**

Transcription factors (TFs) affect the production of mRNAs. In essence, the TFs form a large computational network that controls many aspects of cellular function. This article introduces a computational method to optimize TF networks. The method extends recent advances in artificial neural network optimization. In a simple example, computational optimization discovers a four-dimensional TF network that maintains a circadian rhythm over many days, successfully buffering strong stochastic perturbations in molecular dynamics and entraining to an external day–night signal that randomly turns on and off at intervals of several days. This work highlights the similar challenges in understanding how computational TF and neural networks gain information and improve performance.

## 1. Introduction

Transcription factors (TFs) influence mRNA production. Multiple TFs form inputs into a biochemical network that affects mRNA outputs. The TF networks govern the biochemical dynamics that control much of cellular function.

TF networks pose five key challenges. How do natural processes design TF networks? What TF network architectures commonly arise? What consequences follow from the particular architectures? How can human-engineered TF networks achieve particular design goals? How do TF networks compare with other input-output networks, such as artificial neural networks?

This article introduces a computational optimization method to design TF network models, with focus on optimizing differential equations for the temporal trajectories of TF dynamics. The computational process differs from how natural selection designs actual biochemical TF networks but does provide insight into design by blind search.

On the technical side, the computational approach arises from the great recent advances in automatic differentiation algorithms [1,2,3]. Automatic differentiation provided the key step that transformed modern AI by allowing realistic optimization of large artificial neural networks [4]. In the same way, it is now possible to optimize TF networks that depend on large numbers of parameters.

As often happens with novel computational optimization applications, the techniques from other fields do not work immediately without additional technical modifications and advances. This article introduces several small but essential technical steps. Such steps include expanding the thermodynamically motivated TF input-output function into a computational function for arbitrary network sizes, developing an inverse computational map from realistically motivated parameter bounds to computationally useful parameter ranges for optimization, and finding the various computational hyperparameters and initial conditions that allow successful optimization.

This article also illustrates the method with a simple example and lists promising directions for future work. Given the wide interest in such computational models and the potential for broad future development and insight, this first small step may motivate further progress.

Literature search did not turn up any prior methods for the general optimization of differential equation models for TF dynamics and temporal trajectories of expression. Several studies have applied machine learning techniques and automatic differentiation to analyze differential equation models of cellular biochemistry, focusing on steady-state outcomes [5,6,7,8]. Lopatkin and Collins [9] reviewed various related modeling approaches for microbial biology and their future potential.

Much recent computational work focuses on the general optimization of temporal trajectories arising from differential equation models [10], which motivated my application to TF networks and biochemical dynamics. That computational work also provides methods for optimizing stochastic differential equations, which I illustrate in this article.

## 2. Materials and Methods

I wrote the computer code in the Julia programming language [11]. I used the SciML packages to optimize differential Equations [10]. Efficient optimization depends on automatic differentiation [1,2], which is built into the SciML system. The source code for this article provides the details for all calculations and plotting of results [12]. The following sections highlight aspects of the computations.

### 2.1. Number of Parameters

For *n* TFs, there are 4n rate parameters from the differential equations in Equation (Equation 1). I included the 2n initial conditions as parameters to be optimized, which improves the search process. In the activation function *f*, the *n* TFs potentially bind to *n* different promoters, adding $n^{2}$ values for each of *k* and *h*. There are $n2^{n}$ values of α, and $n(2^{n}-\left(n+1\right))$ values of *r*. Simplifying, the total number of parameters is $n(5+n+2^{n+1})$. For example, n=4 associates with 164 parameters.

As *n* rises above 5, the number of parameters quickly approaches the approximation $n2^{n+1}$. For n=10, there are approximately $2\times 10^{4}$ parameters, and for n=20, there are approximately $4\times 10^{7}$. The large number of parameters favors study of the two-step optimization approach suggested in the Discussion.

### 2.2. Biological Bounds on Parameters

I set bounds on all parameters to be roughly compatible with realistic value ranges [13]. When a parameter concerned the abundance of a molecule, I used number per cell. I list rates m,δ,s,γ on a per second basis. The parameter ranges are
$$\begin{array}{cc} m\;\mathrm{and}\;\delta & =(10^{-4},10^{-2})\\ s\;\mathrm{and}\;\gamma & =(10^{-3},1)\\ k & =(10^{2},10^{4})\\ h & =(0,5)\\ \alpha & =(0,1)\\ r & =(0,10). \end{array}$$

The rate parameters set bounds on the molecular abundances, with mRNA molecules approximately bounded on $\left(0,10^{2}\right)$ and TF protein molecules approximately bounded on $\left(0,10^{3}\right)$.

### 2.3. Optimization of Bounded Parameters

To keep the parameters bounded when using an intrinsically unbounded optimization procedure, I used an algorithmic transformation. The parameter vector used for automatic differentiation and optimization was unbounded. Before feeding the parameters into the differential equations of Equation (Equation 1), I transformed the unbounded values into the bounds of the prior section. For example, to transform an unbounded parameter value, *p*, into the range (0,1), I used a sigmoid function
$$\sigma \left(p,d\right)=\left\{\begin{array}{cc} p<d & k_{1}({1+e^{-10p})}^{-1}\\ p>1-d & ({1+e^{-10p+k_{2}})}^{-1}\\ \mathrm{otherwise}\; & p \end{array}\right.$$
with $k_{1}=d\left(1+e^{-10d}\right)$, and $k_{2}=10\left(1-d\right)+\log\left(d/\left(1-d\right)\right)$, for d=0.01. For the transformed parameter, θ, on the range (0,1), we obtain a biologically meaningful range, (a,a+b) by a+bθ, which, for a=0 or b≫a, is approximately (a,b).

### 2.4. Initial Parameters and Hyperparameters

I chose parameters within their bounded range and then inverted the above transformation to store those parameters on the unbounded scale for optimization. The success of optimization depended on the initial choice of parameters, including the initial numbers of mRNAs and TFs.

In a typical run, I set the initial parameters and then multiplied each value by 1+z for the Gaussian random variable *z*, with mean 0 and standard deviation typically in the range (0.1,0.5). Common initial parameters are *m* from a uniform random variate on $\left(10^{-4},10^{-2}\right)$, $\delta =1.01\times 10^{-4}$, $s=1.01\times 10^{-1}$, $\gamma =1.01\times 10^{-3}$, $k=5\times 10^{2}$, h=2, α from a random uniform variate on (0,1), and r=1. The computer code shows the exact details of all values and calculations.

Setting the decay rates δ and γ near their minimum seemed particularly important for successful optimization. If these values were initially high, the numbers of mRNAs and TFs often quickly decayed toward zero, providing little opportunity for discovering good parameter values.

Optimization success depends on many additional choices, traditionally called *hyperparameters*. For example, I used the Adam algorithm for updating parameter values given the gradient of performance with respect to the parameters [14]. That learning algorithm has several hyperparameters that determine how new parameters are chosen. I typically used a learning rate of 0.002, and reduced that rate when attempting to refine a potentially good solution. I also began the initial optimization search with only a short period of the temporal target trajectory, and then slowly lengthened the fitting period [15]. The computer code shows the full details for these and other choices [12].

### 2.5. Stochastic Fluctuations Vary with Abundance

I set stochastic fluctuations for a molecule with abundance *z* as $\sqrt{z}dW$ in which *W* is a standard normal variate with mean 0 and standard deviation 1, thus $\sqrt{z}W$ has a standard deviation of $\sqrt{z}$. As *z* drops, the ratio of the standard deviation relative to the mean increases. To prevent fluctuations becoming too large relative to the abundance, which can cause negative abundance values in the numerical analysis, for z≤16, I replaced $\sqrt{z}$ with z/4.

## 3. Results

### 3.1. Dynamics of TF Networks

The derivatives with respect to time for numbers of mRNA molecules, *x*, and the TFs produced by those mRNAs, *y*, are
(1)$$\begin{array}{c} \begin{array}{cc} {\dot{x}}_{i} & =m_{i}f_{i}\left(y\right)-\delta_{i}x_{i}\\ {\dot{y}}_{i} & =s_{i}x_{i}-\gamma_{i}y_{i}, \end{array} \end{array}$$
for $m_{i}$ as maximum mRNA production rate, $\delta_{i}$ as the mRNA decay rate, $s_{i}$ as the TF production rate per mRNA, and $\gamma_{i}$ as the decay rate of i=1,⋯,n TFs [16].

In this system, all TFs can potentially modulate the expression of all mRNAs. I found it useful in this initial study to analyze this maximally connected network. As the system becomes optimized, some of the connections may become weak or absent, allowing study of how optimization prunes network connectivity. The computer code provides various options to limit the number of mRNAs influenced by each TF. Thus, one could compare networks that are fully connected networks with networks that are constrained.

### 3.2. TF Network as Input-Output Function

The function $f_{i}$ transforms the numbers of TFs in the vector y into the production level of each mRNA, varying between 0 for complete repression and 1 for maximum production. The activation function arises from thermodynamic theory [17], leading to the calculation for a single TF as [16,18]
$$f\left(y\right)=\frac{\alpha_{0}+\alpha_{1}v}{1+v},$$
in which $v={\left(y/k\right)}^{h}$ is activation by TF abundance *y* relative to the TF–promoter binding dissociation constant, *k*, reshaped by the Hill coefficient, *h*. The parameters $\alpha_{0}$ and $\alpha_{1}$ weight the expression levels when the promoter is unbound or bound, respectively, with α varying between 0 and 1. For two TFs,
$$f\left(y_{1},y_{2}\right)=\frac{\alpha_{0}+\alpha_{1}v_{1}+\alpha_{2}v_{2}+\alpha_{3}rv_{1}v_{2}}{1+v_{1}+v_{2}+rv_{1}v_{2}},$$
in which each TF has its own intensity parameters, $v_{i}={\left(y_{i}/k_{i}\right)}^{h_{i}}$, and *r* quantifies synergy between TFs. My computer code expands *f* for any number of TFs.

### 3.3. Maintaining Circadian Rhythm as a Design Challenge

To illustrate the optimization method, the design goal is for TF 1 to follow a 24 h period. TF abundance above $10^{3}$ molecules per cell corresponds to an “on” state for daytime. Below that threshold, the cell is in an “off” nighttime state.

For a system with n=4 TFs, the differential equations for the system have 164 parameters (see Methods). In Figure 1a, we seek a parameter combination that minimizes the loss measured as the distance between the target circadian pattern shown in the gold curve and the transformed abundance of TF 1 in the green curve. In particular, the daily target rhythm follows (2)$$z_{t}=\frac{\sin(2\pi t+\pi )+1}{2},$$ with *t* in days and passage across 0.5 corresponding to transitions between day and night. The loss is $$L=\sum_{t}({\frac{y_{t}^{2}}{10^{3}+y_{t}^{2}}-\frac{z_{t}^{2}}{0.5+z_{t}^{2}})}^{2},$$ in which $y_{t}$ is the abundance of TF 1, and the transformations of *y* and *z* are Hill functions that normalize values to make the different scales comparable. For the summation, the time values start at t=0 and increment by 0.02 until the end of time for a particular analysis. The values for *y* and *z* in Figure 1 are normalized to the scaling of the plots, as described in the figure legend.

Optimization of the deterministic system in Equation (Equation 1) often finds a nearly perfect fit between the system’s temporal trajectory and the target circadian pattern. Repeated computer runs with different random initialization and search components typically converge to different TF networks.

### 3.4. Stochastic Molecular Dynamics

Different optimized fits for the deterministic system in Equation (Equation 1) had widely varying sensitivities to perturbation by stochastic molecular dynamics. The real challenge is to optimize a stochastic system. To each derivative in Equation (Equation 1), I added a Gaussian noise process weighted by the square root of the molecular abundance. The updated dynamics fluctuate stochastically.

### 3.5. Random External Light Signal for Entrainment

A stochastic system inevitably diverges from the target circadian trajectory. The system may use an external entrainment signal, such as daylight, to correct deviations. In this example, I added a strong boost to the production rate of TF 2 in proportion to the intensity of external light. In particular, the rate of change in TF 2 abundance in the presence of the external light signal is augmented by $10^{6z_{t}-4}$, in which $z_{t}$ is the daily rhythm in Equation (Equation 2).

Initially, the light signal is absent. The signal switches on and off randomly. In Figure 1b, the gold curve shows the external light signal, which switches on in the middle of the third day and stays on for the remaining days shown. The blue curve traces the abundance of TF 2. The updated challenge is for TF 1 to track the circadian pattern, with stochastic molecular dynamics and a randomly occurring external entrainment signal.

### 3.6. Dynamics of an Optimized System

Figure 1 shows the best-performing system obtained by optimization. In panel (a), the system’s trajectory (green) lags the target pattern (gold) during the first few days because of stochastic perturbations from molecular dynamics. When the external light signal switches on in the middle of day 3, the system quickly entrains to the circadian pattern and remains tightly synchronized for the remaining days shown. Panels (c) and (d) show the other two TFs, and panels (e–h) show the mRNAs for each matching TF.

Each stochastic trajectory of the system differs because of the stochastic molecular dynamics and the random switching of the external light signal. Panel (i) shows 20 system trajectories over 20 days, in which the average waiting time between the switching on and off of the light signal is w=2 days. In this case, the signal comes on often enough for the system to correct most deviations caused by stochastic dynamics.

In panel (j), the average waiting time for the light switch is w=1000. Because the light starts in the off state, it essentially never comes on. Thus, the trajectories show how well the system can maintain a circadian pattern in response to internal stochastic molecular perturbations, with no external signal to correct deviations. In this case, most trajectories remain remarkably close to their target over a long time period.

Figure 2a shows the deviations of system trajectories from the circadian target for 1000 sample trajectories. The numbers associated with the *w* labels show the average waiting time between random switches of the external light signal. In each vertical set of circle and lines, the circle shows the median deviation in hours between the target and system entry times into the daylight state. The upper line traces the range of the 75th to 95th percentile, and the lower line traces the 5th to 25th percentile. The left, middle, and right set for each *w* value shows the distribution of deviations at day 10, 20, and 30, respectively. With an entrainment signal of w≤16, the system typically remains close to its target.

When there is effectively no external entrainment, for w=1000, the system inevitably diverges from its target with an increase in the day of measurement. Nonetheless, the match remains very good given the highly stochastic molecular dynamics.

This system performs better than other optimization runs in my study. When I tried to improve the performance of this system further with additional optimization steps and altered optimization hyperparameters, the performance always decreased. The reason remains an open puzzle. Further study may provide insight into the geometry of the performance surface, an analogy for the commonly discussed fitness landscape problems of evolutionary theory [19,20].

### 3.7. TF Logic of an Optimized System

Figure 2b illustrates the TF network input-output function, *f*. This subsection links the observed TF logic to the design challenge of circadian pattern with a randomly fluctuating daylight entrainment signal.

The plots show the generation rate multiplier for the mRNA that makes TF 1. In each plot, the bottom axes show TF protein numbers, labeled as p1 and p2 for TFs 1 and 2, respectively. The scale is log10(1+y), in which *y* is the number of TF molecules per cell. Rows show increasing amounts of TF protein 3, labeled p3. Columns show increasing amounts of TF 4.

A high value of p2 associates with a strong external light signal, as in Figure 1b. The TF network only strongly stimulates p1 production when the external light signal is strong and both p3 and p4 are high. From the plots in Figure 1b–d, those conditions are only met from a couple of hours past the temporal transition into daytime through midday. Transition into daytime is marked by the vertical dotted lines in panels (a) and (b). Those requirements allow the system to entrain accurately to the external daylight signal when it is present.

In the absence of an external light signal, p2 rises to a lower level at the onset of daylight. Once again, with high levels for both p3 and p4, the rise of p2 increases expression of the mRNA that produces p1. That pattern creates the same temporal entrainment to daylight, but in this case solely by internal signals from the cell’s intrinsic dynamics. However, when there is no external light signal, the system lacks the very strong rise in expression of p1 in midday that seems to be the main entrainment force to an external daylight signal.

In this way, the cell entrains relatively weakly to its stochastic and less reliable internal circadian signals and entrains relatively strongly to an external daylight signal, when that external signal is present. The use of two internal signals, p3 and p4, may help to buffer the effects of stochastic perturbations.

The story outlined here gives a plausible interpretation of the system design. However, when I tried to further optimize this system, I observed significant reductions in performance. That decay raises a puzzle. What causes the sensitivity of the parameter tuning with respect to the dual challenges of buffering stochastic molecular dynamics and entraining to an external light signal when present? In other words, what is the geometry of the optimization surface with respect to the parameters?

Finally, Figure 2b suggests that the TF logic tends to create steep sigmoid step changes in response to changes in TF input concentrations. That pattern matches a common theoretical assumption that TF logic can be modeled by a piecewise continuous pattern [21]. Further optimization studies may show the conditions that favor such piecewise continuous outputs versus other patterns.

## 4. Discussion

### 4.1. Optimize a Neural Network and Fit a TF Network

A TF network is a function, *f*, that inputs *n* TFs and outputs *N* mRNA expression levels. In the current work, I used the currently favored thermodynamic model for TF binding to promoters to calculate the TF input-output function. Alternatively, one could take a two-step approach in optimization modeling. First, use an artificial neural network model for the function, *f*. Optimize the system’s temporal trajectory with respect to a design challenge. Second, fit the TF parameters against the optimized neural network function, *f*.

This two-step approach would allow one to use highly efficient neural network algorithms for the initial optimization. Then, in the fitting of the TF parameters to the optimized neural network function, one could explore the role of various biochemical constraints on the TFs and their effects. One may also gain insight into the geometry of optimization surfaces for TFs versus other types of computational networks.

The current neural network literature is actively exploring how and why various network architectures succeed or fail in gaining information and improving performance [22,23,24,25]. Can we bring TF networks and cellular computations for information processing and control within this broader conceptual framework? Future work on TF optimization modeling will play an important role in answering this question.

### 4.2. Large Networks, Flat Fitness Surfaces, and Genetic Variation

Greater dimensionality of computational networks and more parameters sometimes lead to better optimization. The reasons are not fully understood [25]. It may be that more dimensions and parameters tend to smooth the optimization surface, perhaps also flattening the surface in many directions. With respect to TF networks, larger systems may adapt better to new challenges [26]. In more highly parameterized and flatter optimization surfaces, a particular TF variation would have less average effect.

In a flatter optimization surface and fitness landscape, genetic variants and mutational effects may tend to be smaller. Smaller fitness effects associate with more genetic variation. There are some hints that, in theory, flatter landscapes and more genetic variation associate with increased heritability of failure [27,28,29]. Thus, studying TF optimization models may lead to better understanding of fitness landscapes and genetic variation.

## Figures and Tables

**Figure 1 biology-11-01294-f001:**
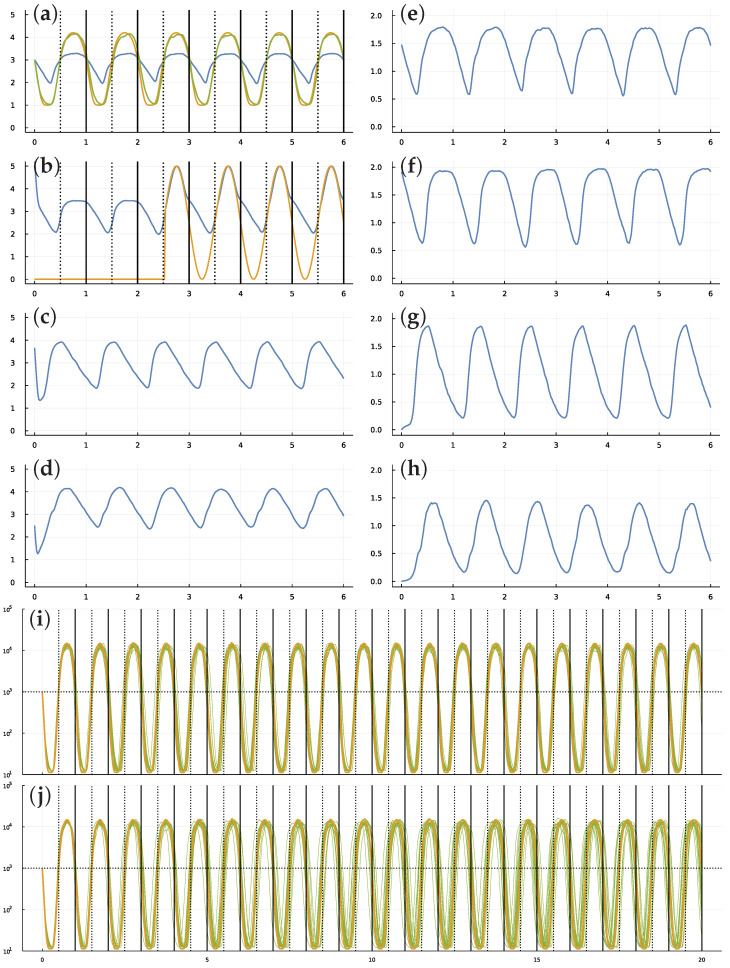
Circadian dynamics with stochastic fluctuations and random daylight signal. Stochastic dynamics of TF proteins (**a**–**d**) and the mRNAs that produce them (**e**–**h**) over six days. The parameters were obtained from the best result of all optimization runs, in which *best* means the closest match of cellular dynamics to a circadian pattern, as defined in the following paragraphs. The vertical lines in (**a**,**b**) show entry into daylight (dotted) and nighttime (solid). The *y*-axis is log10(1+y) for number of molecules per cell, *y*. The optimization design goal is for the blue curve in (**a**), the number of TF 1 molecules, to match a circadian rhythm. To define the optimization loss value to be minimized, the number of TF 1 molecules, *y*, is transformed by a Hill function, $\tilde{y}=y^{2}/\left(1000^{2}+y^{2}\right)$, to yield the green curve, which traces values $1+4\tilde{y}$. The gold curve traces the target circadian pattern. The optimization loss value to be minimized is the sum of the squared deviations between the gold and green curves at 50 equally spaced time points per day. The number of TF 2 proteins in (**b**) is influenced by the internal cellular dynamics and is also increased in response to an external daylight signal (see text). The availability of the light signal switches on and off randomly. It is initially off. The average waiting time for a random switch in the presence or absence of the signal is *w*, measured in days. In this example, w=2. The signal turns on around sunrise of day 3 and stays on for the remaining days shown. Because the switching is random, daylight can be present or absent for several days in a row, or it can switch on and off several times in one day. In this particular example, looking at the match between cellular state shown by green curve in (**a**) compared with the target gold signal, stochastic molecular perturbations push the cellular rhythm behind the actual circadian pattern during the first few days. When the daylight signal appears in the middle of day 3, the system entrains to the external signal and closely matches the target circadian pattern for the remaining days shown in the plot. Panels (**i**,**j**) illustrate the match of internal cellular state (green) and target circadian pattern (gold) over 20 days. Each plot shows a sample of 20 stochastic trajectories for cellular state, showing the magnitude and the randomness in the degree of mismatch between actual and target trajectories. In (**i**), the average waiting time between random switching of for the presence or absence of the external light signal is w=2 days. In (**j**), the waiting time is w=1000 days. Because the signal starts in the off state, in (**j**) the external signal essentially never comes on. Thus, the green stochastic cellular dynamics in (**j**) illustrate the ability of the cell to hold a circadian rhythm over many days in the absence of an external light signal for entrainment.

**Figure 2 biology-11-01294-f002:**
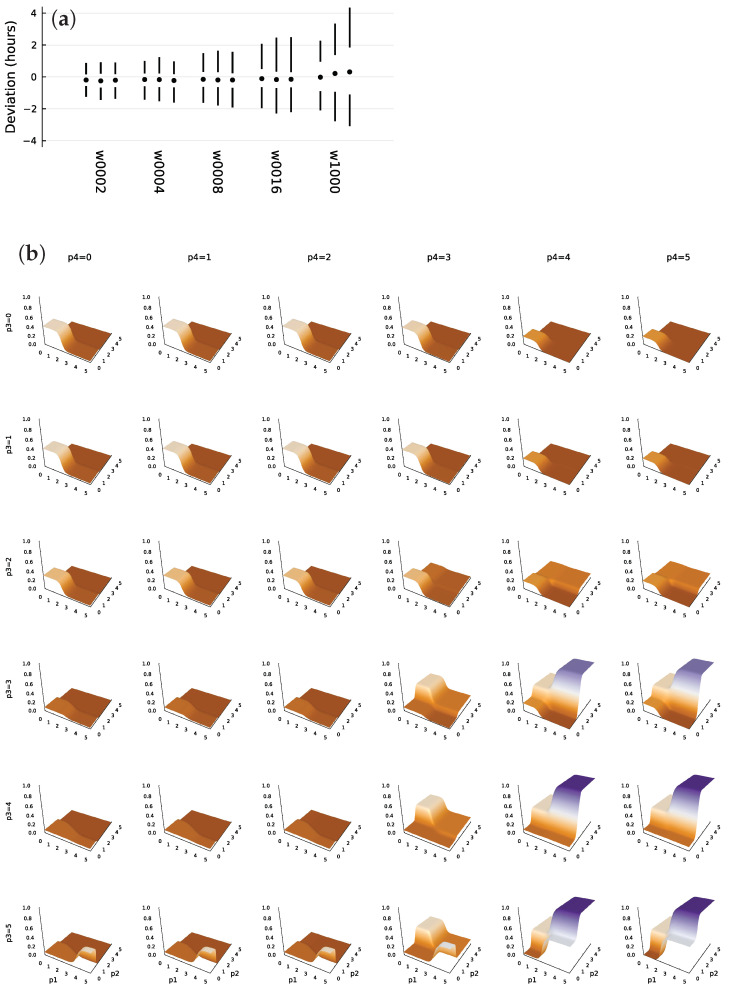
Stochastic perturbations to entrainment (**a**) and TF logic for mRNA expression (**b**). Panel (**a**) presents the deviation of cellular dynamics from the circadian target pattern. Each set of two vertical lines and a circle show the distribution of the deviation between the entry into daytime cellular state and the actual onset of daytime. The circle denotes the median of 1000 stochastic cellular trajectories. The upper line shows the 75th percentile at the bottom and the 95th percentile at the top. The lower line shows the 5th (bottom) and 25th (top) percentiles. Each set of three distributions shows, from left to right, the distribution of deviations measured at 10, 20, and 30 days. The w labels below denote the average waiting time in days for random switches between the presence or absence of the external daylight entrainment signal (see caption for Figure 1). From left to right, the waiting times vary over 2,4,8,16,1000. Shorter waiting times provide more frequent entrainment signals, reducing the consequences of the intrinsic molecular stochasticity of cellular dynamics. For w1000, the external signal essentially never occurs, thus the set of distributions shows the intrinsic cellular stochasticity and the ability of the cell to maintain a circadian pattern in the absence of an external signal. TF protein levels control mRNA expression. The four TF proteins form the inputs, and the four expression levels for the associated mRNAs form the outputs. This panel shows all four TF inputs and the associated mRNA expression level for protein 1 as the surface levels of each plot. In each plot, the basal axes quantify the levels of TF 1 and TF 2, labeled as p1 and p2 in the bottom row of plots. The scale is log10(1+y) for TF protein level *y*. The height of each plot shows the relative expression level triggered by the TF inputs, scaled from 0 for complete repression to 1 for maximum expression. The rows from top to bottom show increasing levels of TF 3, labeled as p3. The columns from left to right show increasing levels of TF 4, labeled as p4. See the text for interpretation of the plots.

## Data Availability

All computer code, parameters and output used to generate the figures are available on Zenodo [12].

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
