# Peer review of "Optimization of Transcription Factor Genetic Circuits"

_biology, 2022, doi:10.3390/biology11091294_

Round 1
Reviewer 1 Report
The author has presented a computational study on the design of a 4-node TF network. The study is interdisciplinary and interesting on many levels with results that set the stage for further exploration. This reviewer has the following concerns:
(1) The TF network considered is an all-to-all network (though this is not explicitly stated). i.e, TF <-> mRNA for all TFs and mRNAs. Is this a realistic architecture in biology -- as an illustrative example it seems a bit ad hoc. A discussion of this may be helpful.
(2) The author uses a novel two-step optimization process to fit the parameters of the TF network. In the first step, an artificial neural network formalism is used to map the function relating the TF inputs to the mRNA outputs. This step would be greatly aided if the author could present the ANN being optimized with a figure and the weights to be determined. Since this ANN appears to be a shallow network, it is not clear how many parameters could be optimized with this network, and how these parameters relate to the problem at hand. A visualization might again clarify a lot of these ideas.
(3) What is the exact function used for constructing the target trajectory? Also a closed-form expression for the loss function would be helpful -- the author provides a narrative in the figure caption.
(4) Since the data seems to be a time series, are the instances constructed on a daily basis? Are there enough instances to complete the fitting? Is there a risk of overfitting the ANN model? These are some questions that remain. Too many parameters could lead to overfitting and the symptoms described by the author that the optimization process degrades with more steps. A flatter fitness surface could again be traced to overfitting, and it is necessary to explore this possibility out before advancing other hypotheses.
(5) In the section on "Random external light signal for entrainment", the author mentions that he added a "stong boost to the production rate of TF 2 in proportion to the intensity of external light." Though the expression appears apparent, it would be helpful if the author could present the new equation for TF 2 production rate.
(6) The author concludes, ".. flatter landscapes and more genetic variation associate with increased heritability of failure". What is the meaning of failure here -- loss of network function? It would be a stretch to relate this to disease yet.
Answers to the above questions would be helpful to the reader. This reviewer feels that the manuscript may be presented with more detail to be self-contained.
Reviewer 2 Report
This manuscript authored by Steven Frank introduces a computational framework to optimize TF networks.
The title should be modified to be more descriptive and reflective of the work performed to inform readers.
Some places in the manuscript can be benefitted with re-phrasing. For instance, the last sentence in Abstract needs revision. in the first sentence in abstract, an alternative way is to say “production” of mRNAs instead of expression. On page 5, fourth paragraph following the heading of TF logic of an optimized system, ‘production’ of the mRNA is more suitable than ‘expression.” In the second and third paragraphs in introduction, natural processes do not ‘design’ TF networks. TF networks were likely evolved. An option is to change ‘design’ to ‘give rise to’ for both ‘design(s)’
Based on the introduction, it is unclear what and if there were prior studies similar to this one. There need more references throughout introduction. The relevance of this work and the impact it can make for experimental biologists or broader readership is unclear.
Round 2
Reviewer 1 Report
The author has sufficiently addressed all the concerns raised by this reviewer and clarified the relevant passages in the manuscript. The manuscript in its current form does not provide evidence link with disease and hence the conclusion in this regard is something the author could reconsider (though the reviewer appreciates the additional supporting references for heritability changes). With regard to the accompanying software, it is suggested that the author annotate each folder & script in the software to enhance its usability by any interested scientist. As it stands, given the complexity of the challenge considered, the utility level is not as great as it could be.
Author Response
I removed all mention of disease. I added further details about the software to the README.md file, including a summary of each script file. That README file also contains a description of the key directories in the software. I uploaded the revisions to the Zenodo archive for the software.